# Analysis of Factors Affecting Creep of Wood–Plastic Composites

**Li Feng * and Weiren Xie**

College of Mechanical and Electrical Engineering, Northeast Forestry University, Harbin 150040, China; wang_lifeng100@nefu.edu.cn

* Correspondence: fengli86@163.com; Tel./Fax: +86-451-82190397

**Abstract:** Wood–plastic composite (WPC) materials are mainly used as flooring in buildings or as structural load-bearing plates, and will undergo creep deformation during use, resulting in structural failure and safety problems. Therefore, this work adopted the orthogonal test method to carry out creep tests on wood–plastic composites. We used the range method and variance analysis method to process the creep data and analyze the influence of the load, temperature, and relative humidity on the creep strain in specimens of wood–plastic composites. The results showed that the creep strain of the WPC specimens changed significantly with a change in the load stress, while a change in relative humidity had no significant effect on the creep strain. When the relative humidity was increased from 55% to 65%, the creep strain increased by 0.03%, but when the temperature was increased from 30 °C to 35 °C, there was no significant difference in the creep strain. However, when the temperature was increased from 30 °C to 40 °C and from 35 °C to 40 °C, a significant difference in the creep strain of the WPC specimens was observed.

**Keywords:** wood–plastic composite; creep; temperature; humidity; load





## 1. Introduction

Wood–plastic composite (WPC) is a recycled material that is mainly composed of biomass fibers and a certain proportion of plastic. WPC is formed after special processing. It is mainly used in the fields of construction, outdoor facilities, and decorative materials. It has become an ideal environmental protection material for residential and commercial purposes. The WPC studied in this work is used as a bearing material in the construction industry for planking, bridge slabs, and so on. Because of the creep characteristics of WPCs, they will undergo a large creep deformation, posing the risk of structural failure and safety problems.

The creep behavior of wood–plastic composites has a direct influence on their mechanical properties. Since materials have a higher stiffness at low temperature, the creep resistance of wood–plastic composites is greater at low temperature [1]. The creep behavior of unmodified and functionally modified thermoplastic wood–plastic composites in a certain temperature range has been studied using short-term flexural creep tests, and it was found that when the temperature rose above room temperature, the degree of the creep of wood–plastic composites increased with increasing temperature, and both the instantaneous deformation and creep rate increased [2–7].

When WPCs are used under various external conditions, their performance is difficult to predict due to the changes in the environmental factors such as temperature [8]. Studies of the influence of the outdoor natural climate conditions on the flexural and creep properties of WPCs showed that these properties are degraded after exposure to the sun for 2–6 months, so that the strain produced by the material is higher than that at room temperature after 6 months; then, due to the decrease in the temperature, humidity, and ultraviolet intensity, the strain generated by the material is smaller than that at room

temperature [9]. In addition to the temperature, humidity also affects the creep behavior, so that an increase in the humidity will accelerate the creep of WPCs [10].

The residual tensile strength, creep strain, and partial deflection of WPCs were measured with a short-term bending creep test to study the effect of the wood flour content on the creep resistance and bending properties of WPCs. The results showed that with the increase in wood flour content, the creep modulus of WPCs increases, the creep strain decreases, and the bending creep resistance increases gradually [11–14]. A study of the mechanical properties and creep resistance of bamboo-fiber-reinforced regenerated poly-lactic acid composites (BFRPCs) showed that the BFRPCs with 60% fiber content had the best creep resistance. When the fiber content was greater than 70%, the creep resistance decreased [15]. The effect of wood flour ethanoylation on the creep properties of wood flour/polypropylene composites was studied, and it was found that the bending strength and tensile strength of wood flour/polypropylene composites increased with the increase in the mass fraction of the wood flour ethanoylation particles. When the mass fraction of wood flour ethanoylation particles reached 13%, the creep resistance of the composites reached the maximum value [16].

The creep and fatigue properties of WPCs under bending and cyclic deformation were tested. The results showed that the creep resistance of WPCs was poor when the size (thickness) was too small, and the creep resistance of WPCs improved with increased size and core number [17,18].

The influence of the loading stress on the flexural creep of WPC was studied through creep tests. The results showed that the short-term flexural creep performance was strongly related to the stress level, and the creep velocity of WPC increased with increasing stress [19–21]. At the beginning of the test, the stress had little effect on the bending deformation growth rate of WPC. However, with an increasing stress level, the strain increase rate of the material increased gradually [22].

To date, some research has been carried out on the creep of wood–plastic composites both domestically and abroad. In practical applications, creep often occurs in the natural environment. Temperature, humidity, the wood flour content, constant stress, and the types of wood flour and plastic will affect the creep and therefore the quality of wood–plastic composites. Currently, the studies reported in the literature cover only a small range of the possible effects on the creep behavior and quality of WPCs. For example, research on the effect of the temperature on the properties of wood–plastic composites mainly focuses on high temperatures. Therefore, in this work, poplar wood powder and polyethylene were used as the main raw materials, and the orthogonal test method was used to design the proportions of the raw materials and the processing parameters in the preparation of WPCs. The bending strength of WPCs was evaluated by the three-point bending loading test. Then, the orthogonal test method was adopted to carry out the creep test of the wood–plastic composites, and the range method and variance analysis method were used to process the creep data and analyze the influence of the load, temperature, and relative humidity on the creep strain of wood–plastic composites. These results were highly significant for guiding the production of WPCs and improving WPC quality.

## 2. Material Preparation Test

In this work, polyethylene PE50d012 (5000S) and poplar powder (PP) were used as the main raw materials for the specimen preparation. The PE was produced by Daqing Petrochemical, with a density of 0.954 g/cm$^3$ and a melt index of 0.9 g/10 min. The PP was supplied in the form of homopolymer pellets. The poplar powder was produced by Qingdao Fujilin New Energy Co., Ltd. (Qingdao, China), with a particle size of 80 mesh. The coupling agent was CMG9804, which is a graft HDPE compatibilizer produced by Nantong Ri Zhi Sheng Polymer New Material Technology Co., Ltd. (Nantong, China).

In this work, the effects of molding temperature, screw speed, wood–plastic ratio, coupling agent content, and granulation temperature on the properties of a WPC were

mainly studied. The technological parameters of the specimen preparation were designed using the orthogonal test method, as shown in Table 1.

**Table 1.** Orthogonal test table for specimen preparation.

| Test No | Molding Temp (°C) | Screw Speed (r·min⁻¹) | Wood–Plastic Ratio | Coupling Agent | Granulation Temp (°C) | Bending Strength (MPa) |
|---|---|---|---|---|---|---|
| A1 | 1(150) | 1(30) | 1(55:45) | 1(2%) | 1(150) | 55.19 |
| A2 | 1(150) | 2(50) | 2(60:40) | 2(3%) | 2(160) | 58.25 |
| A3 | 1(150) | 3(70) | 3(65:35) | 3(4%) | 3(170) | 58.95 |
| A4 | 1(150) | 4(90) | 4(70:30) | 4(5%) | 4(180) | 66.71 |
| B1 | 2(160) | 1(30) | 2(60:40) | 3(4%) | 4(180) | 58.05 |
| B2 | 2(160) | 2(50) | 1(55:45) | 4(5%) | 3(170) | 58.45 |
| B3 | 2(160) | 3(70) | 4(70:30) | 1(2%) | 2(160) | 59.55 |
| B4 | 2(160) | 4(90) | 3(65:35) | 2(3%) | 1(150) | 62.06 |
| C1 | 3(170) | 1(30) | 3(65:35) | 4(5%) | 2(160) | 63.97 |
| C2 | 3(170) | 2(50) | 4(70:30) | 3(4%) | 1(150) | 62.45 |
| C3 | 3(170) | 3(70) | 1(55:45) | 2(3%) | 4(180) | 58.43 |
| C4 | 3(170) | 4(90) | 2(60:40) | 1(2%) | 3(170) | 53.47 |
| D1 | 4(180) | 1(30) | 4(70:30) | 2(3%) | 3(170) | 51.35 |
| D2 | 4(180) | 2(50) | 3(65:35) | 1(2%) | 4(180) | 60.14 |
| D3 | 4(180) | 3(70) | 2(60:40) | 4(5%) | 1(150) | 60.16 |
| D4 | 4(180) | 4(90) | 1(55:45) | 3(4%) | 2(160) | 57.33 |

The wood flour was dried at 80 °C in an electric constant-temperature drying oven for 12 h, until the moisture content was lower than 2%. The wood flour, PE, and coupling agent were mixed in a high-speed mixer for 10 min, and then extruded in a single-screw extruder. The size of the forming port was 40 mm × 4 mm, as shown in Figure 1. According to the ASTM D 790 standard [23], the specimens were cut into samples with dimensions of 80 mm × 13 mm × 4 mm. The test was divided into 16 groups, with 6 test specimens in each group. Before mixing, the wood–plastic ratios were 55:45, 60:40, 65:35, and 70:30. The content of the coupling agent was 2%, 3%, 4%, and 5% based on the total weight of wood flour and PE. During specimen preparation, the granulation temperature and molding temperature were 150 °C, 160 °C, 170 °C, and 180 °C. The screw speeds were 30, 50, 70 and 90 rpm.

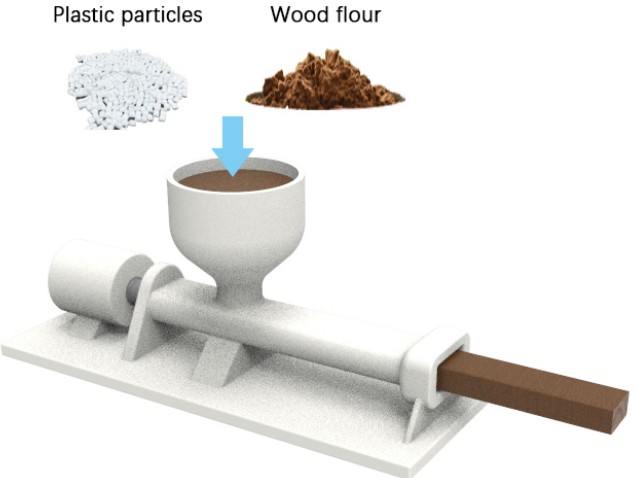

**Figure 1.** The extrusion process.

According to the relevant requirements of the ASTM D790 standard, a three-point bending load test was carried out on the prepared specimens, as shown in Figure 2, with a span of 64 mm and an indenter speed of 2 mm/s. Five specimens were tested in each group, and the average value was taken. The bending load was recorded and the bending strength

of each group was calculated. The results are shown in Table 1, and provide reference data for the creep test of the WPC.

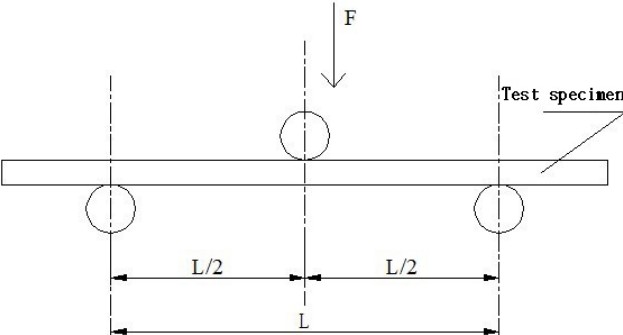

**Figure 2.** Three-point bending load test.

## 3. Creep Test

In this work, the creep test mainly studied the influence of the load, temperature, and relative humidity on the creep of the WPC. The creep testing device was divided into three parts: bench, loading device, and data acquisition [24]. According to the structural requirements for the span, loading rod diameter, support diameter, and data acquisition in ASTM D 7031 [25], the design was based on the basic structure of the universal mechanical testing machine. A schematic diagram of the device is shown in Figure 3.

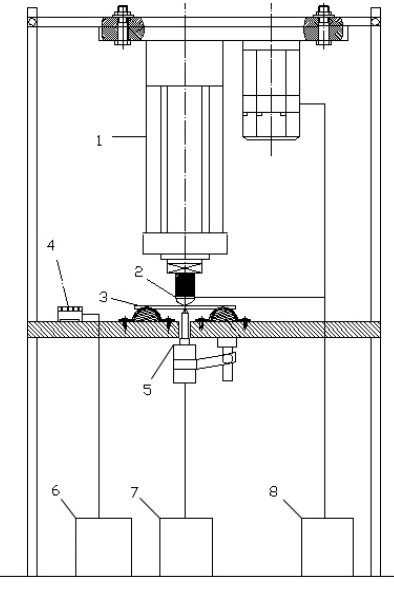

**Figure 3.** Schematic diagram of creep testing device: 1—electric cylinder; 2—pressure Sensor; 3—specimen; 4—temperature and humidity sensor; 5—linear displacement sensor; 6—temperature and humidity controller; 7—acquisition card; 8—control system.

Group B2 specimens with a moderate bending strength (Table 1) were selected for the creep test. The creep test was designed using the orthogonal test method, and the three-point bending load method was adopted to load the specimens. The load was 40%, 50%, or 60% of the bending strength of Group B2 in shown in Table 1. When the set load value was increased, the load remained constant, did not increase over time. The span of the test piece was 64 mm. The temperature and humidity values of the test piece environment were set according to the data shown in Table 2. When the temperature and humidity were lower than the set value, the test device automatically adjusted them to

ensure a constant temperature and humidity. The creep of five specimens was tested in each group, and the average value was taken as the creep value of the group. The test time for each specimen was 24 h, and the data were recorded every 1 min.

**Table 2.** Creep test data.

| Test No. | Load (MPa) | Temperature (°C) | Humidity (%) | Strain (%) |
|---|---|---|---|---|
| 1 | 1(50% of stress) | 1(30) | 1(45) | 1.76 |
| 2 | 1(50% of stress) | 2(35) | 2(55) | 1.94 |
| 3 | 1(50% of stress) | 3(40) | 3(65) | 2.45 |
| 4 | 2(40% of stress) | 1(30) | 3(65) | 1.02 |
| 5 | 2(40% of stress) | 2(35) | 1(45) | 1.28 |
| 6 | 2(40% of stress) | 3(40) | 2(55) | 1.67 |
| 7 | 3(60% of stress) | 1(30) | 2(55) | 2.21 |
| 8 | 3(60% of stress) | 2(35) | 3(65) | 2.44 |
| 9 | 3(60% of stress) | 3(40) | 1(45) | 3.28 |

## 4. Results and Discussion

### 4.1. Creep Test Analysis

Through the creep test, it was found that the bending creep of the WPC produced different deformations under different loads, temperatures, and humidity conditions. Using the creep-test data, taking time in minutes as the abscissa and the creep-deformation variable as the ordinate, a creep-deformation scatter diagram was plotted using Excel software. Since the trends of the creep-deformation diagram for the test data of each group were generally consistent, the creep-test data obtained from Test No. 7 shown in Table 2 were selected to draw the creep- deformation diagram, as shown in Figure 4. It can be seen in Figure 4 that the creep deformation increased gradually with increasing creep time. At the initial stage of constant load, the creep deformation increased sharply; in the middle stage of creep, the creep deformation increased rapidly; as time continues to increase, the creep deformation increased slowly; and in the subsequent time of the test stage, the creep variable did not increase for two or three hours, and the increase rate was zero.

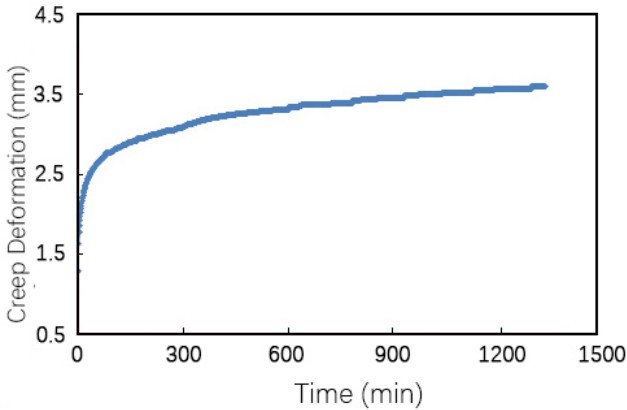

**Figure 4.** Creep-deformation diagram.

The span L of the test piece was 64 mm, and the thickness h of the test piece was 4 mm. The creep-strain value was calculated using the strain formula $\varepsilon = \frac{6h}{L^2} \times f$, as shown in Table 2. The range-analysis method was adopted to analyze the data in Table 2, and the analysis results are shown in Table 3.

**Table 3.** Creep strain range analysis.

|  | Load (MPa) | Temperature (°C) | Humidity (%) |
|---|---|---|---|
| $Q_1$ | 6.15 | 4.99 | 6.32 |
| $Q_2$ | 3.97 | 5.66 | 5.82 |
| $Q_3$ | 7.83 | 7.4 | 5.91 |
| $q_1$ | 2.05 | 1.66 | 2.11 |
| $q_2$ | 1.32 | 1.89 | 1.94 |
| $q_3$ | 2.64 | 2.47 | 1.97 |
| r | 1.32 | 0.81 | 0.17 |

Note: $Q_i$ represents the sum of the creep strain of the load, temperature, and humidity factors in the test at a certain level i, where $i = 1, 2, 3$; $q_i = \frac{Q_I}{3}$; and r is the range. In any column, $r = \max\{q_1, q_2, q_3\} - \min\{q_1, q_2, q_3\}$.

According to the data presented in Table 3, the maximum r value of the load column was 1.32, indicating that the load had the greatest influence on the creep of the WPC, followed by the r value of the temperature column, and a minimum r value of the humidity column of only 0.17, indicating that humidity had the least influence on the WPC. The influence of these three factors on the creep of the wood–plastic composite materials in descending order was: load, temperature, and humidity. Theoretically, the combination of conditions for maximum bending creep of the WPC material is a load of 60% of the stress, a temperature of 40 °C, and humidity of 45%.

The variances shown in Table 2 were analyzed, and the results are shown in Table 4.

**Table 4.** Test of the intersubjectivity effect.

|  | Type III Sum of Squares | df (Freedom) | Mean Square | F | Sig. (Significance Level) |
|---|---|---|---|---|---|
| Calibration Model | 3.701 [a] | 6 | 0.617 | 47.905 | 0.021 |
| Intercept | 36.200 | 1 | 36.200 | 2811.066 | 0.000 |
| Load | 2.622 | 2 | 1.311 | 101.822 | 0.010 |
| Temperature | 1.032 | 2 | 0.516 | 40.054 | 0.024 |
| Humidity | 0.047 | 2 | 0.024 | 1.839 | 0.352 |
| Deviation | 0.026 | 2 | 0.013 |  |  |
| Total | 39.927 | 9 |  |  |  |
| Corrected Total | 3.727 | 8 |  |  |  |

Note: [a]. $R^2 = 0.993$ (adjusted to $R^2 = 0.972$).

It was observed in the results of the test of the intersubjectivity effect shown in Table 4 that the calibration model column, F = 47.905, Sig. (=0.021) < $\alpha$ (=0.05), showing that the model was effective for these creep test statistics. In the stress column, F = 101.822, Sig. = 0.010 < 0.05, indicating that the change of the load applied to the wood–plastic composite material had a significant effect on the flexural creep. In the temperature column, F = 40.054, Sig. = 0.024 < 0.05, indicating that the change in external temperature also had a significant effect on the creep of the WPC. In the humidity column, F = 1.839, Sig. = 0.352 > 0.05, indicating that the change in external humidity had no evident effect on the bending creep of the WPC. The value of $R^2$ was 0.993, indicating that 99.3% of the variation in the creep strain could be explained by the model. The adjusted value of $R^2$ was 0.972, indicating that 97.2% of the variation in the creep strain could be explained by the model. A comparison of the $Rv^2$ values before and after the adjustment showed that the model could explain at least 97% of the variation of the creep strain, again showing that the model had a good fit for this statistical result. Table 4 also shows that the effects of the three factors on the creep of the WPC was in the order of load, temperature, and humidity, which was the same as the results obtained when analyzing the test data with the range analysis method.

The S-N-K method was used to compare the load and temperature, load and humidity, and temperature and humidity creep strain under different levels of average, with the results presented in Table 5. There are three columns in the stress subset column of Table 5. The absence of a similar subset means that there were significant differences in the creep strain between stresses of 40, 50, and 60%. There are only two columns in the temperature subset column of Table 5. This shows that there were similar subsets. For the temperatures of 30 and 35 °C, and the corresponding creep strain set at a significance level of 0.05, there was no significant difference. For the temperatures of 30 and 40 °C, the creep strain of the WPC was significantly different than at 35 and 40 °C.

**Table 5.** Pairwise comparison S-N-K test.

| Load | N | Subset | | | Temperature | N | Subset | | Humidity | N | Subset |
|------|---|--------|------|-------|-------------|---|--------|-------|----------|---|--------|
| | | 1 | 2 | 3 | | | 1 | 2 | | | 1 |
| 40% | 3 | 1.323 | | | 30 °C | 3 | 1.663 | | 55% | 3 | 1.94 |
| 50% | 3 | | 2.05 | | 35 °C | 3 | 1.887 | | 65% | 3 | 1.97 |
| 60% | 3 | | | 2.643 | 40 °C | 3 | | 2.467 | 45% | 3 | 2.107 |
| Sig. | | 1 | 1 | 1 | Sig. | | 0.137 | 1 | Sig. | | 360 |

*4.2. Analysis of Influencing Factors of Creep Performance*

4.2.1. Effect of Temperature on Creep Strain

According to the test data, the creep strain trends for each specimen at constant loads of 40, 50, and 60% and temperatures of 30, 35 and 40 °C were plotted, as shown in Figure 5. It can be observed in Figure 5 that the creep rate changed from high to low, and remained essentially unchanged. This behavior was in accordance with the typical creep trend chart. It can be observed in Figure 5a that the creep strain increased with increasing temperature. When the load was 40% of the stress, the creep strain increased from 1.02% at 30 °C to 1.67% at 40 °C; when the load was 40% of the stress, the creep strain increased with longer time. When the time was between 0 and 60 min, the creep rate increased rapidly. Creep strain increased by 0.05% at 30 °C, by 0.06% at 35 °C, and by 0.1% at 40 °C. The creep strain was stable after 120 min. The creep strain of the WPC increased with increasing temperature. This was mainly because the test temperature was below the melting point of the material, so there was no crystallization behavior. With the increase in the temperature, the thermal movement of molecules increased, the resistance of the matrix to the deformation decreased, the distance between molecules expanded, the friction between molecules decreased, and the creep strain rate of the WPC increased.

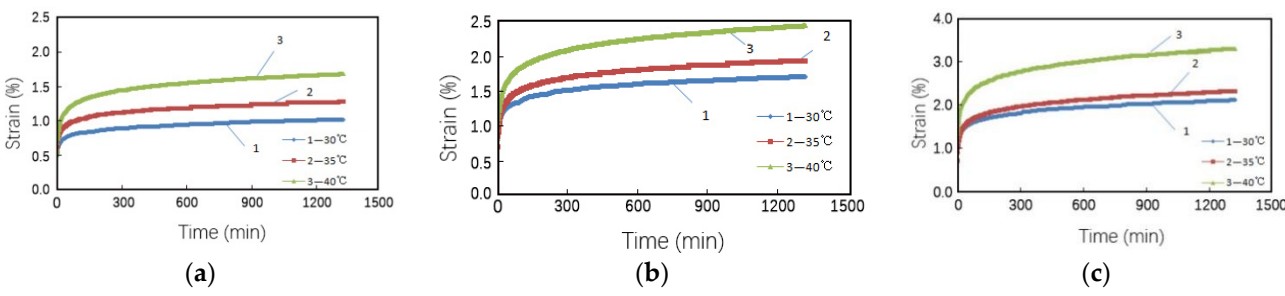

**Figure 5.** Effect of the temperature on the creep strain with load change: (**a**) constant load of 40% of the stress; (**b**) constant load of 50% of the stress; (**c**) constant load of 60% of the stress.

It can be observed in Figure 5c that when the external temperature was 30 or 35 °C, the creep strain curves of the composite material were close to each other with only small differences, which was completely consistent with previous studies [26]. When the load was 40% of the stress, the difference between the creep strains was 0.26%. When the load was 50% of the stress, the difference between the creep strains was 0.18%. When the load

was 60% of the stress, the difference between them was 0.25%. Therefore, under constant load, the creep strain of wood plastic composite was not significantly different at external temperatures of 30 and 35 °C, which was consistent with the analysis results for 30 and 35 °C presented in Table 5.

### 4.2.2. Effect of Load on Creep Strain

According to the 24 h creep test data, the creep strain trends of the WPC under the conditions of 30, 35 and 40 °C and loads of 40, 50, and 60% of the stress were plotted (Figure 6). It can be observed in Figure 6 that the creep strain increased with longer time. The creep rate changed from high to low, and then the creep rate remained unchanged, conforming to the typical creep trend. It can be observed in Figure 6b that the creep strain increased rapidly between 0 and 60 min and then slowed down between 60 and 120 min; when the test time reached 2 h, the creep rate became stable. When the temperature was kept constant and the specimen was subjected to different constant loads, the creep rate increased with increasing load in the first 60 min before the creep strain was generated. At 30 °C and within 60 min of creep strain, when the load was 40% of the stress, the creep strain was 0.29%; when the load was 50% of the stress, the creep strain was 0.49%, and when the load was 60% of the stress, the creep strain was 0.93%.

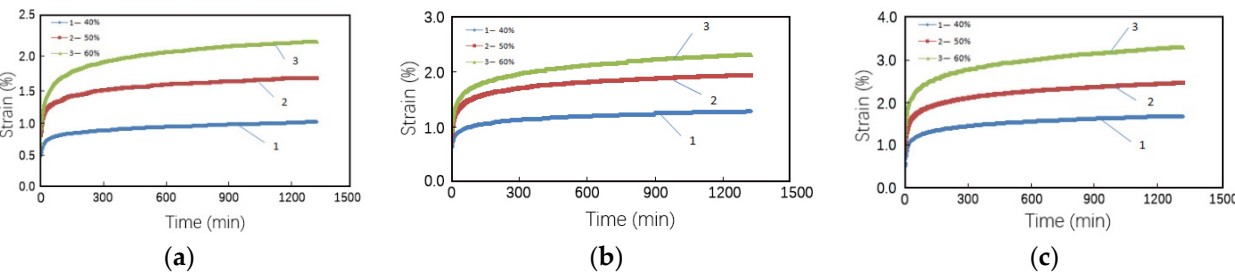

**Figure 6.** Creep strain trend for different temperatures: (**a**) 30 °C; (**b**) 35 °C; (**c**) 40 °C.

According to Figure 6a, when the temperature was 30 °C, the load increased from 40% to 60% of the stress, the creep strain of the specimen increased from 1.02% to 2.21%, the stress increased by 20%, and the creep strain doubled. According to Figure 6b, when the temperature was 35 °C, the load increases from 40% to 60% of the stress, the creep strain of the specimen increased from 1.28% to 2.44%, the stress increased by 20%, and the creep strain was nearly doubled. According to Figure 6c, when the temperature was 40 °C, the load increased from 40% to 60% of the stress, the creep strain of the specimen increased from 1.67% to 3.28%, the stress increased by 20%, and the creep strain was nearly doubled.

For a fixed temperature, the creep strain increased with a constant load applied to the specimen. A similar result was reported in a previous study, in which the 24 h short-term creep test of a Masson pine/HDPE composite showed that the load was 10% of the stress and the creep strain was 0.11%; the load was 20% of the stress and the creep strain was 0.38%; and the load was 30% of the stress and the creep strain was 0.4%. A 24 h short-term creep test of Chinese fir/HDPE composite showed that the load was 10% of the stress and the creep strain was 0.26%; the load was 20% stress and the creep strain was 0.51%; and the load was 30% of the stress and the creep strain was 0.55% [27].

### 4.2.3. Effect of Humidity on Creep Strain

According to Table 3, when the humidity was 45%, 55%, and 65%, the average value of the creep strain was 2.11%, 1.94%, and 1.97%, respectively. The influence trend of humidity on the creep strain of the WPC was plotted according to Table 3, as shown in Figure 7. It can be observed in Figure 7 that when the relative humidity was greater than 45%, the creep strain increased with increasing relative humidity; when the relative humidity increased from 55% to 65%, the creep strain increased from 1.94% to 1.97%, and the increase range was 0.03%, which was consistent with previous studies [28]. An examination of the data

presented in Table 3 showed that the range of the humidity term (r = 0.17) was much lower than the range of the stress term (r = 1.32) and the range of the temperature term (r = 0.87), indicating that the change in the humidity had the least influence on the creep strain of the WPC. It can be observed in Table 4 that Sig. = 0.352 in the humidity column was much larger than Sig. = 0.05, also indicating that the influence of humidity on the creep strain of the WPC was not significant. It can be observed in Table 5 that there was only one subset of humidity, also showing that the change of relative humidity between 45%, 55%, and 65% did not lead to a significant difference in the creep strain.

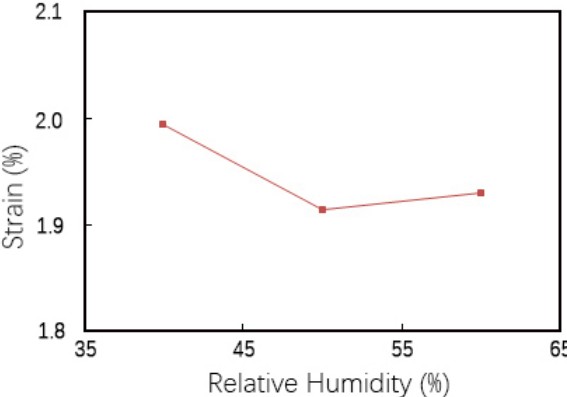

**Figure 7.** Effect of humidity on the creep strain of the WPC.

The WPC contained a nonpolar molecular polymer, and the wood powder was dispersed between the composites and wrapped by the polymer, leading to a significant reduction of the contact area between the wood powder and water. Therefore, the water absorption of the WPC was quite poor, so that the change in relative humidity did not have a significant effect on the creep strain of the WPC.

## 5. Conclusions

In this work, a creep test device for WPCs was designed to carry out creep tests. The range method and variance analysis method were used to analyze the effects of the load stress, temperature, and relative humidity on the creep properties of a WPC. The results showed that: (1) the creep strain of the WPC increased significantly with increasing load stress at the same temperature while under the same load conditions; (2) the creep strain of the WPC increased with increasing temperature; (3) the influence of the load on creep was greater than that of temperature, and the influence of humidity on the creep strain of the WPC was not significant. The results of this work could be useful for guiding the production of WPCs and improving WPC quality.

In this work, only 24 h short-term creep strain was examined, while long-term creep tests were not carried out. In future work, to better study the creep deformation of WPCs, a mechanical model and mathematical model will be used to predict the creep strain of WPCs. By analyzing the quality of the fits of different models to the creep data for WPCs, the creep strain of WPCs over long durations will be predicted.

**Author Contributions:** Conceptualization, L.F. and W.X.; methodology, L.F.; software, W.X.; validation, W.X.; formal analysis, W.X.; investigation, L.F.; resources, L.F.; data curation, W.X.; writing—original draft preparation, L.F.; writing—review and editing, L.F.; visualization, W.X.; supervision, L.F.; project administration, L.F.; funding acquisition, L.F. All authors have read and agreed to the published version of the manuscript.

**Funding:** Natural Science Foundation of Heilong Jiang province (c201346); The Fundamental Research Funds for the Central Universities (2572018BF07).

**Acknowledgments:** This project was supported by the Natural Science Foundation of Heilong Jiang province (c201346) and the Fundamental Research Funds for the Central Universities (2572018BF07).

**Conflicts of Interest:** The authors declare no conflict of interest.

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
