# Peer review of "Analysis of Factors Affecting Creep of Wood–Plastic Composites"

_forests, doi:10.3390/f12091146_

Round 1
Reviewer 1 Report
State of art contains resources from 24 literary sources, which I consider sufficient.
In the state of art, I would be asked for more information about the specific use of WPC boards (in relation to the researched issues). The part describing the sample preparation is processed sufficiently with a detailed description of the process and composition of WPC samples.
The results are adequate processed in the form of Tables and graphical dependencies.
Conclusion could be made in bullets for fast recognition of article goals and results.
Author Response
Point 1: State of art contains resources from 24 literary sources, which I consider sufficient.
Response 1: Thank you for comments.
Point 2: In the state of art, I would be asked for more information about the specific use of WPC boards (in relation to the researched issues). The part describing the sample preparation is processed sufficiently with a detailed description of the process and composition of WPC samples.
Response 2: Thanks for your suggestions.More information about the specific use of WPC boards are added in line 21-26.The part describing the sample preparation has been modified in line 89-109.
Point 3: The results are adequate processed in the form of Tables and graphical dependencies.
Response 3: Thank you for comments.
Point 4:Conclusion could be made in bullets for fast recognition of article goals and results.
Response 4: Thanks for your suggestions. Conclusion has been made in bullets in line 295-299.

Reviewer 2 Report
Dear Authors,
I have read the article entitled „ Analysis of factors affecting creep of Wood-Plastic Composites".
This is a straightforward study. In this work, poplar wood powder and polyethylene were used as the main raw materials for WPC. The influence of several factors on the creep strain of WPC have been analyzed. The results of this work can be significant for guiding the production of WPC and improving the WPC quality.
The subject of the study falls within the scope of the Journal. The title and abstract reflect the content. The introduction initiates the reader on the subject and a well-selected list of references was used. The methodology was clearly explained. All figures and tables are mentioned within the text.
I have only a few observations.
Page 1 lines 34-40: the text is too long and it is difficult to follow, please split it.
Page 3 line 104: please specify how many groups and samples were used?
Page 5 line 137: comparisons with the findings from other studies may be added, if possible
Page 9 line 283: please add the sentence from page 2 line 85 „The results of this work can be useful for guiding the production of WPC and improving the WPC quality” to the conclusions.
Page 9 line 292: please add to the Reference list all the standards used for the tests.
I hope my revision was of help.
Author Response
Point 1: Page 1 lines 34-40: the text is too long and it is difficult to follow, please split it.
Response 1: Thanks for your suggestions. This part has been modified in line 39-44.
Point 2: Page 3 line 104: please specify how many groups and samples were used?
Response 2:Thanks for your suggestions.The test is divided into 16 groups with 6 test specimens in each group. This part has been specified in line105-106.
Point 3: Page 5 line 137: comparisons with the findings from other studies may be added, if possible
Response 3:Thanks for your suggestions. Comparisons with the findings from other studies has been added in line 223, line 255-262, line 275.
Point 4: Page 9 line 283: please add the sentence from page 2 line 85 “The results of this work can be useful for guiding the production of WPC and improving the WPC quality” to the conclusions.
Response 4:Thanks for your suggestions. The sentence “The results of this work can be useful for guiding the production of WPC and improving the WPC quality” has been added in line 299-300.
Point 5: Page 9 line 292: please add to the Reference list all the standards used for the tests.
Response 5: Thanks for your suggestions. All the standards used for the tests are added to the Reference list in line 359-360, line 363-364.
